# Epstein–Barr Virus LMP1 Induces Soluble PD-L1 in Nasopharyngeal Carcinoma

**DOI:** 10.3390/microorganisms9030603

**Published:** 2021-03-15

**Authors:** Kina Kase, Satoru Kondo, Naohiro Wakisaka, Hirotomo Dochi, Harue Mizokami, Eiji Kobayashi, Makoto Kano, Takeshi Komori, Nobuyuki Hirai, Takayoshi Ueno, Yosuke Nakanishi, Miyako Hatano, Kazuhira Endo, Makiko Moriyama-Kita, Hisashi Sugimoto, Tomokazu Yoshizaki

**Affiliations:** Division of Otolaryngology and Head and Neck Surgery, Graduate School of Medical Science, Kanazawa University, Kanazawa 920-8640, Japan; kina@med.kanazawa-u.ac.jp (K.K.); wakisaka@med.kanazawa-u.ac.jp (N.W.); h_dochi@med.kanazawa-u.ac.jp (H.D.); haruesun@med.kanazawa-u.ac.jp (H.M.); e_kobayashi@med.kanazawa-u.ac.jp (E.K.); makoto-kano@med.kanazawa-u.ac.jp (M.K.); takkomori@med.kanazawa-u.ac.jp (T.K.); NHhira@med.kanazawa-u.ac.jp (N.H.); uenotaka@med.kanazawa-u.ac.jp (T.U.); nakanish@med.kanazawa-u.ac.jp (Y.N.); mhatano@med.kanazawa-u.ac.jp (M.H.); endok@med.kanazawa-u.ac.jp (K.E.); mkita@med.kanazawa-u.ac.jp (M.M.-K.); sugimohi@med.kanazawa-u.ac.jp (H.S.); tomoy@med.kanazawa-u.ac.jp (T.Y.)

**Keywords:** EBV, sPD-L1, PD-L1, LMP1, nasopharyngeal carcinoma

## Abstract

Nasopharyngeal carcinoma (NPC) is an Epstein–Barr virus (EBV)-associated malignancy. The principal oncogene of EBV, latent membrane protein 1 (LMP1), induces the expression of programmed death-ligand 1 (PD-L1), which is an immunosuppressive transmembrane protein and a promising therapeutic target for various malignancies. Recent studies have revealed an association between the level of soluble PD-L1 (sPD-L1) and disease progression. However, the role of sPD-L1 in NPC or its relevance to LMP1 has not been elucidated. This study aimed to examine whether LMP1 induces sPD-L1 in vitro and analyze the clinical relevance of LMP1, PD-L1, and sPD-L1 in NPC patients. Analysis of nasopharyngeal cell lines revealed that LMP1 induces both cellular PD-L1 and sPD-L1. Analysis of biopsy specimens from 32 NPC patients revealed that LMP1 expression was significantly correlated with PD-L1 expression. Finally, the serum sPD-L1 level in NPC patients was higher than that in the controls. Moreover, the sPD-L1 level in the advanced stage was higher than that in the early stage. However, LMP1 expression, PD-L1 expression, and sPD-L1 levels were not associated with prognosis. These results suggest that LMP1 induces both sPD-L1 and PD-L1, which are associated with NPC progression.

## 1. Introduction

Epstein–Barr virus (EBV) is strongly associated with the etiology of nasopharyngeal carcinoma (NPC) [1,2]. Latent membrane protein 1 (LMP1), a principal oncogene of EBV [3], contributes to the promotion of invasion and metastasis as well as initial oncogenesis of NPC [4]. Moreover, LMP1 promotes escape from various immune recognition processes, especially cytotoxic T lymphocyte response that kills tumor cells and virus-infected cells [5]. Programmed death ligand 1 (PD-L1), a transmembrane glycoprotein, binds to PD-1 on T lymphocytes and suppresses T lymphocyte activation [6]. Tumor cells express PD-L1 and utilize the PD-L1/PD-1 checkpoint for immune evasion [7]. Fang et al. reported that LMP1 induced cellular PD-L1 expression in vitro, and their data confirmed that blocking both LMP1 and PD-L1 expression is a potential therapeutic approach for NPC [8]. Moreover, Yang et al. showed that a high level of soluble PD-L1 (sPD-L1) is significantly associated with advanced-stage NPC [9]. In addition, sPD-L1 is a prognostic biomarker of malignant melanoma [10]. Asanuma et al. reported that high sPD-L1 levels are associated with a poor prognosis in soft tissue sarcomas, and their findings suggested that sPD-L1 is released by aggressive tumors [11]. Thus, the level of sPD-L1 is clinically significant in certain malignancies. Taken together, it is noteworthy to investigate the relationship of LMP1 with PD-L1 and sPD-L1 in NPC. In this study, we examined the role of LMP1 in the production of sPD-L1 and the clinical relevance of LMP1 expression, PD-L1 expression, and sPD-L1 levels in NPC.

## 2. Materials and Methods

### 2.1. Patients Characteristics

This study included 35 patients with primary NPC who were treated at Kanazawa University Hospital between July 2007 and July 2019. This study was approved by the Kanazawa University ethics committee (IRB#2016-033). All patients provided written informed consent.

### 2.2. Cell Lines

Human immortalized nasopharyngeal cell lines, NP69T cells, and NP69T-LMP1 cells (NP69T cells transfected with pLNSX-LMP1 and stably expressing LMP1), were a kind gift from Dr. George Sai Wah Tsao (University of Hong Kong) [12]. These cell lines were cultured in keratinocyte serum-free medium (Thermo Fisher Science, Kyoto, Japan).

### 2.3. Immunohistochemical Analysis

The expression of PD-L1 and LMP1 was immunohistochemically examined in 32 of 35 primary NPC biopsy specimens obtained during the first medical examination. All patients did not receive any treatments before the biopsies and all samples were not derived from a tumor recurrence. PD-L1 expression was assessed by both tumor proportion score (TPS) and combined positive score (CPS). TPS is the percentage of PD-L1-positive tumor cells in the total number of viable tumor cells. CPS is the percentage of PD-L1-positive cells (tumor cells, lymphocytes, and macrophages) in the total number of viable tumor cells [13]. PD-L1 and LMP1 expression were independently evaluated by two investigators (K. K. and S. K.) who were blinded to the clinical data of the patient. Three-micrometer sections were cut from paraffin blocks of a primary lesion. Paraffin sections were deparaffinized, treated with 3% hydrogen peroxide, and incubated with a protein blocker (Dako, Glostrup, Denmark). Next, the sections were incubated at 4 °C overnight with anti-PD-L1 antibody (ab205921, 1:500, Abcam, Cambridge, MA, USA) or anti-LMP1 antibody (M0897, 1:100, Dako, Glostrup, Denmark). After washing with PBS, the sections were exposed to EnVision+ secondary antibody (Dako, Glostrup, Denmark). Then, the sections were counterstained with hematoxylin. Furthermore, 7 of 32 primary NPC specimens were used for dual fluorescence immunostaining of PD-L1 and LMP1. Paraffin sections were deparaffinized, treated with 3% hydrogen peroxide, and incubated with a protein blocker (Dako, Glostrup, Denmark). The sections were incubated at 4 °C overnight with primary antibodies. Next, the sections were exposed to goat anti-mouse Alexa Fluor 594 and anti-rabbit Alexa Fluor 488 IgG secondary antibodies (1:500, Thermo Fisher Science, Kyoto, Japan) after washing with PBS. Then, the sections were counterstained with 4′,6-diamidino-2-phenylindole (DAPI, P36962, Thermo Fisher Scientific, Kyoto, Japan).

### 2.4. Western Blotting

The cells and medium were collected 48 h after seeding the cells in 10 cm dishes. The total cell lysates were denatured in radioimmunoprecipitation assay lysis buffer (Tris-HCI, 1% NaCl, sodium deoxycholate, sodium dodecyl surface [SDS], sodium orthovanadate, NaF, and fluoride) and boiled for 5 min. The medium was concentrated to 50 fold using Vivaspin VS2041 (Sartorius, Goettingen, Germany). The samples were separated using SDS-polyacrylamide gel electrophoresis and transferred to nitrocellulose membranes (Bio-Rad Laboratories, Hercules, CA, USA). The membranes were blocked with 5% skim milk Tris-buffered saline-Tween 20 (TBST) and incubated at 4 °C overnight with the following primary antibodies: PD-L1 (#13684, 1:1000, Cell Signaling Technology, Danvers, MA), LMP1 (M0897, 1:200, Dako, Glostrup, Denmark), and α-tubulin (T9026, 1:5000, Sigma Aldrich, St. Louis, MO, USA). After washing with TBST (Santa Cruz Biotechnology, Santa Cruz, CA, USA), the membranes were exposed to horseradish peroxidase-conjugated secondary antibodies (Bio-Rad Laboratories, Hercules, CA, USA). The signals were detected using enhanced chemiluminescence reagent (GE Healthcare, Tokyo, Japan).

### 2.5. Serum Samples

Serum samples were obtained from 32 patients with primary NPC and 10 controls. Because 3 of the 35 patients with NPC had double cancer (one patient with stomach cancer, one with lung cancer, and one with prostate cancer), we excluded them from this analysis. All patients did not receive any treatments before the serum collection and all samples were not derived from a tumor recurrence. Control samples were obtained from 10 patients who did not have any pathologies that may affect sPD-L1 levels, including carcinomas, infectious disease, and autoimmune diseases at diagnosis. The serum was frozen at −80 °C before further processing.

### 2.6. sPD-L1 Detection Using ELISA

For detecting sPD-L1 in serum samples, PD-L1 ELISA kit (KE00074, Proteintech, Manchester, UK) was used. Serum samples of 32 patients with NPC and 10 controls were analyzed according to the manufacturer’s protocol.

### 2.7. Statistical Analysis

SPSS statistics package ver. 23 (IBM, New York, NY, USA) was used for data analysis. Mann–Whitney U-test was used to analyze the clinical characteristics of the patients. The Spearman rank correlation coefficient was used to analyze the association among LMP1, PD-L1, and sPD-L1. Kaplan–Meier curves were used to examine the overall survival (OS) and progression-free survival (PFS) according to LMP1, PD-L1, and sPD-L1 levels in patients with NPC. We excluded 3 patients who had a double cancer from survival analyses. LMP1 or PD-L1 expression score of 10% and more was considered positive [13], and sPD-L1 levels of 0.1 ng/dL and higher were considered positive [14]. The differences in survival were analyzed using the log-rank test. Cox proportional hazard regression analysis was used to assess univariate and multivariate analysis results of clinicopathological factors such as sex, smoking status, alcohol consumption, TNM classification, clinical-stage, TPS status, CPS status, LMP1 status, and sPD-L1 status. The differences were considered statistically significant when the *p*-value was less than 0.05.

## 3. Results

### 3.1. LMP1 Induces Both Cellular PD-L1 Protein and sPD-L1 in Nasopharyngeal Cells

Fang et al. reported that LMP1 induces cellular PD-L1 expression in NPC cell lines [8]. However, whether LMP1 also induces sPD-L1 remains unknown. Thus, to clarify the association between LMP1 and PD-L1, we examined the cell extracts of the non-malignant EBV-negative nasopharyngeal cell line NP69T. The expression of PD-L1 protein increased in LMP1-expressing NP69T cells (NP69T-LMP1) compared with that in NP69T cells (Figure 1). This result indicates that LMP1 induced the expression of the PD-L1 protein. Finally, to investigate whether the PD-L1 protein was released from nasopharyngeal cells, we performed western blotting with a culture medium. PD-L1 protein was detected in the concentrated medium of NP69-LMP1 cells (Figure 1). Therefore, LMP1 induces the sPD-L1 and PD-L1 in nasopharyngeal cells in vitro.

### 3.2. Expression of PD-L1 and LMP1 Are Correlated in NPC Tissues Samples

The results of the cell culture model analysis suggested that LMP1 promoted both cellular PD-L1 expression and sPD-L1 levels. Therefore, we examined LMP1 and PD-L1 expression in the 32 NPC tissue samples using immunohistochemical analysis. The characteristics of the patients are shown in Table 1. There were 32 men and 3 women, and 8 patients had stages I–II NPC and 27 had stages III–IV NPC. LMP1 expression was observed mainly in the tumor cell membrane, as indicated by brown staining (Figure 2A). In contrast, PD-L1 expression was positive in both tumor and inflammatory cell membranes (Figure 2B). Next, we performed dual fluorescence immunostaining and assessed the expression of LMP1 and PD-L1 in the same tissue sample. There were several cells that expressed both LMP1 and PD-L1 in the NPC tissue sample (Figure 2F). LMP1 expression score was significantly associated with PD-L1 TPS (*p* = 0.023, r = 0.402, Figure 3A). These results support our hypothesis that LMP1 promotes the expression of PD-L1 in NPC tissue samples, as well as in cell culture in vivo.

### 3.3. Relationship between LMP1 and PD-L1 Expression in NPC Tissue Samples

We next assessed the characteristics of the patients with NPC and the LMP1 and PD-L1 status in the tissue samples. The characteristics of patients based on LMP1 and PD-L1 expression (PD-L1 TPS and CPS) are shown in Table 2. There was a significant increase in PD-L1 TPS in advanced-stage patients (*p* = 0.03) compared to that in early-stage patients. In addition, LMP1 expression was higher in females (*p* = 0.009) than in males. This result suggests that the number of PD-L1-positive tumor cells was associated with the NPC stage. Moreover, LMP1 expression differed between males and females.

### 3.4. Elevated Serum sPD-L1 Level in Patients with NPC

Because PD-L1 expression was elevated in NPC tissues, we next investigated the sPD-L1 levels in serum samples. sPD-L1 levels significantly differed between the controls and patients with NPC (*p* = 0.031, Figure 4). In addition, we found a significant increase in sPD-L1 levels in NPC patients with advanced tumor stage (*p* = 0.036, Table 3) compared with that in patients with the early tumor stage. However, sPD-L1 level was not associated with LMP1 expression score (*p* = 0.49, r = 0.13), PD-L1 TPS (*p* = 0.92, r = 0.02), and PD-L1 CPS (*p* = 0.88, r = 0.03) in patients with NPC. Taken together, these results showed that sPD-L1 levels were elevated in patients with NPC. Moreover, high sPD-L1 levels were associated with advanced tumor stage in NPC.

### 3.5. Expression of LMP1, PD-L1, and sPD-L1 Level Was Not Related to Prognosis of Patients with NPC

Finally, we examined whether LMP1 and PD-L1 expression served as a prognostic factor in patients with NPC because both PD-L1 expression and sPD-L1 levels are prognostic factors in other malignancies [10,15,16]. PFS of LMP1-positive patients with NPC (LMP1 expression score: more than 10%) was not significantly different from that of LMP1-negative patients (*p* = 0.965, Figure 5A). Similarly, PD-L1 TPS and CPS and sPD-L1 status (positive or negative) were not prognostic factors (*p* = 0.178, *p* = 0.089, *p* = 0.657, respectively, Figure 5B–D). Univariate analysis showed that smoking and metastasis were significantly associated with PFS, and multivariate analysis showed that both smoking and metastasis were independent variables associated with PFS (Table 4). In addition, there was no significant difference in OS (Appendix A). Univariate analysis showed that sex was significantly associated with OS (Appendix A). These results suggest that neither PD-L1 expression nor sPD-L1 level was a prognostic factor for patients with NPC.

## 4. Discussion

NPC is a virus-related cancer. In particular, the EBV principal oncogene LMP1 plays an important role in the pathogenesis of NPC [17,18]. Some studies have reported that PD-L1 expression is upregulated in virus-associated malignancies, such as EBV-related Hodgkin’s lymphoma and human papilloma-virus related head and neck cancers [19,20]. These studies suggested that induction of PD-L1 expression was associated with immune tolerance in virus-associated tumors and blocking PD-L1 expression was a potential therapeutic approach. There are few treatment options for recurrent NPC [21]. Therefore, blocking PD-1/PD-L1 expression has attracted attention as a new therapeutic approach for NPC [22]. Ma et al. reported that PD-1 blockade has clinical significance for the treatment of recurrent NPC after heavy pretreatment [23]. In addition, a previous study found that LMP1 upregulated PD-L1 expression in NPC cell lines [8]. A similar result was found in a study on natural killer/T cell lymphoma, which showed that LMP1 mediated PD-L1 expression through the MAPK/NF-κB pathway [24]. In this study, we found that there was a slight correlation between LMP1 and PD-L1 expression (Figure 2). Thus, our results were consistent with previous results and confirmed that LMP1 might contribute to the induction of PD-L1 expression in NPC.

In contrast, sPD-L1 has received much attention because some studies revealed its clinical significance in malignancies [25,26,27]. sPD-L1 suppresses T cell activation and induces T cell apoptosis [10,28,29]. These analyses indicated that tumor cells escape the host immune system via circulating sPD-L1 levels. We found that sPD-L1 level was induced by LMP1 in the nasopharyngeal cell lines (Figure 1). To our knowledge, this is the first study to demonstrate an association between LMP1 and sPD-L1 levels. Our findings suggested that both sPD-L1 and PD-L1, which are induced by LMP1, inhibited T cell activation. Similar to most membrane-bound protein soluble forms, an upregulation mechanism of sPD-L1 involves shedding from the surface membrane of PD-L1-positive cells. Chen et al. suggested that matrix metalloproteinase (MMP) produces sPD-L1 from the surface membrane of PD-L1-positive cells [30]. Previously, we reported that LMP1 enhanced the expression of MMP1, MMP3, and MMP9 in NPC [31,32]. Taken together, LMP1 may be associated with the production of sPD-L1 by inducing MMP in NPC. Alternatively, sPD-L1 is related to alternatively spliced variants of PD-L1 [10,33].

Recent studies have reported that PD-L1 expression is associated with factors affecting malignant potential, including immune evasion [34,35,36]. Moreover, Li et al. showed that serum levels of sPD-L1 increase with tumor stage in breast cancer [37]. Similarly, in our study, NPC patients with advanced tumor stage showed elevated sPD-L1 levels (Table 3). In addition, Chang et al. reported that high PD-L1 expression in lung cancers was associated with an advanced clinical stage [38]. Our current results also revealed that higher PD-L1 expression was associated with an advanced stage (PD-L1 TPS, Table 2). These results suggest that the aggressive behavior of certain malignancies could be partially related to high PD-L1 expression as well as high sPD-L1 levels. However, unlike previous reports [24,39], serum sPD-L1 levels were not significantly associated with PD-L1 expression in the tissue samples in our analysis. Some studies reported sPD-L1 as a candidate serum biomarker [10,40,41]. Similarly, our findings showed that pretreatment sPD-L1 levels were higher in NPC patients than in control patients (Figure 4). Unlike previous studies, we found no evidence to support the hypothesis that sPD-L1 levels, LMP1 expression, and PD-L1 expression were associated with prognosis, and only four patients died in this study. Some of the reasons were differences in techniques and protocols between previous studies and our analysis and heterogeneity of PD-L1 expression using immunohistochemistry [42]. In addition, there may be some factors that affect the dissociation of sPD-L1 and PD-L1, such as endemic NPC and non-endemic NPC. Important limitations of this analysis were that the sample size was small and some cases had a short follow-up time after treatment. Future studies should focus on the predictive value of sPD-L1 and PD-L1 for determining the efficacy of immune checkpoint inhibitors required.

In summary, we found that serum sPD-L1 was detectable and there was a significant difference between sPD-L1 levels and tumor stage in NPC. In contrast, sPD-L1 levels were not associated with the outcome of primary treatment because there were few events of death as per OS. However, about 50% of recurrent patients with NPC have distant metastasis, and it is often difficult to obtain a biopsy tumor tissue sample and evaluate PD-L1 expression [43]. sPD-L1 levels could replace tumor PD-L1 expression as a surrogate biomarker. Therefore, further assessment is needed to determine whether sPD-L1 could predict the outcome of treatment with immune checkpoint inhibitors such as PD-1 inhibitors.

## 5. Conclusions

The findings of this study will be useful for a better understanding of the pathophysiology of NPC. Our findings suggest that LMP1 induces sPD-L1 and PD-L1 expression. Further analysis is necessary to apply our findings regarding potential biomarkers in NPC to clinical practice.

## Figures and Tables

**Figure 1 microorganisms-09-00603-f001:**
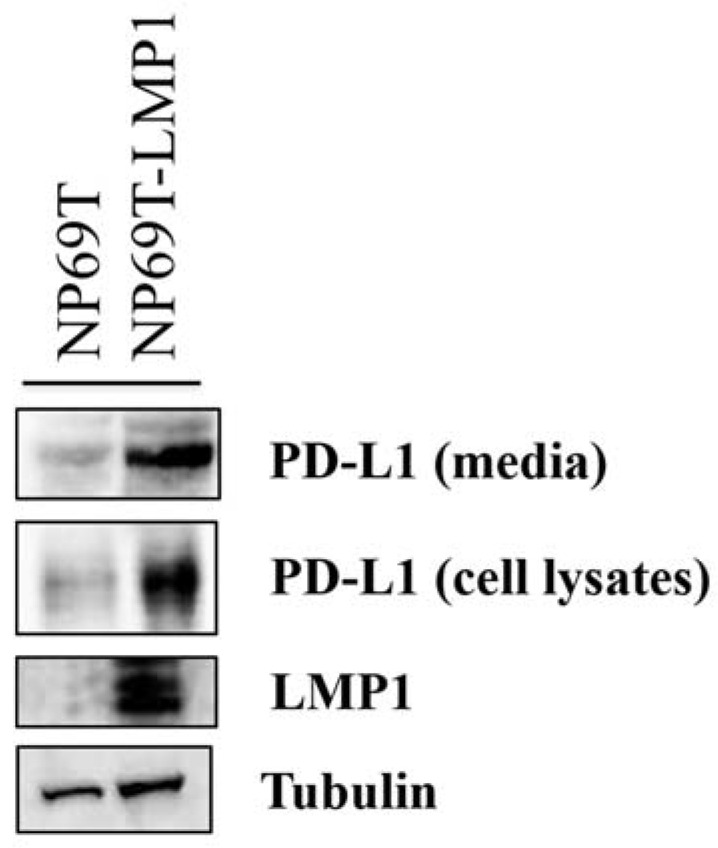
Latent membrane protein 1 (LMP1) induces both cellular programmed death-ligand 1 (PD-L1) protein and soluble PD-L1 (sPD-L1). (Top lane) The media of both NP69T and NP69T-LMP1 cells were analyzed with western blotting for detecting PD-L1 levels. (Second, third, and fourth lane) Western blotting of PD-L1 and LMP1 in NP69T and NP69T-LMP1 cells. Total cell lysates and medium were collected from cell culture at 48 h. The harvested medium was concentrated to 50 fold.

**Figure 2 microorganisms-09-00603-f002:**
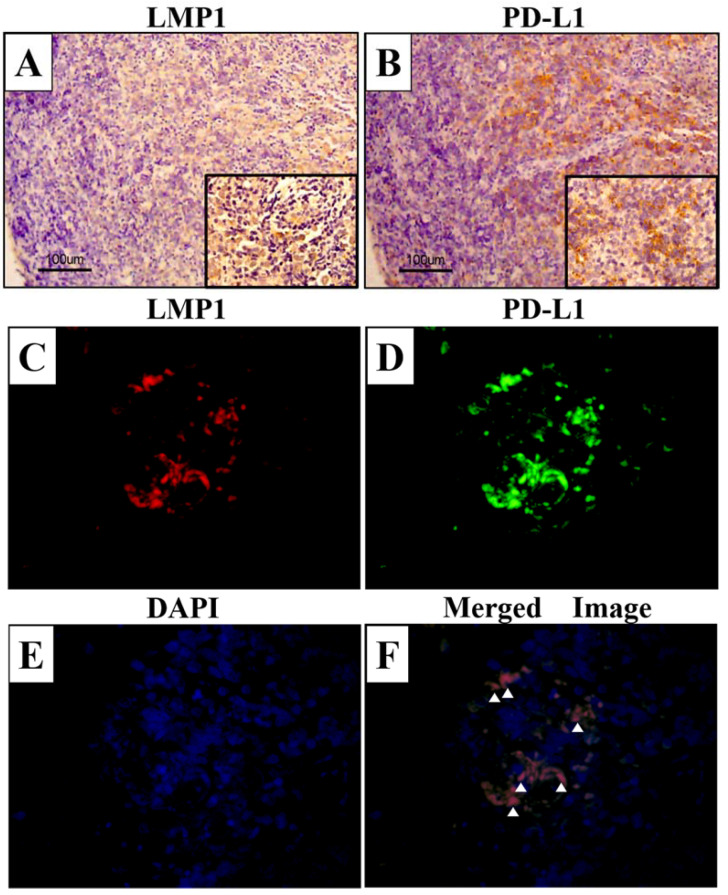
LMP1 and PD-L1 expression in NPC tissue samples. The specimens were immunostained with LMP1 and PD-L1 antibody. Brown staining indicates positive cell membrane expression of LMP1 (**A**) and PD-L1 (**B**). Dual fluorescence immunostaining of LMP1 (**C**) and PD-L1 (**D**), and DAPI (**E**) in NPC tissue samples. (**F**) Merged image of LMP1 and PD-L1. White arrowheads indicate the colocalization of LMP1 and PD-L1. Original magnifications (**A**,**B**): ×100, inset figures: ×400, scale bar: 100 μm. Original magnifications (**C**–**F**): ×600.

**Figure 3 microorganisms-09-00603-f003:**
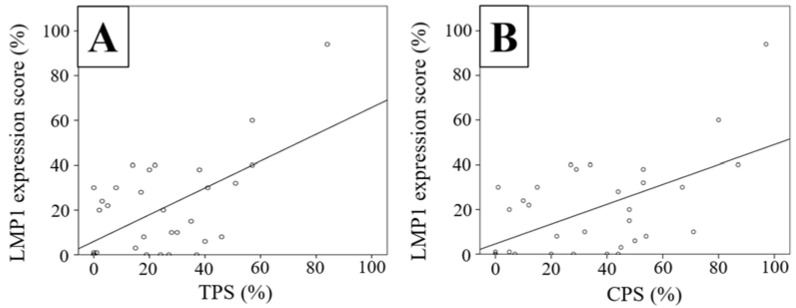
Expression of PD-L1 (TPS and CPS) corelates with LMP1 in NPC tissue samples. Data were analyzed using the Spearman rank correlation coefficient. Statistical significance was defined as a *p*-value less than 0.05. (**A**) Relationship between LMP1 expression score and TPS (*p* = 0.023, r = 0.402). (**B**) Relationship between LMP1 expression score and CPS (*p* = 0.008, r = 0.461).

**Figure 4 microorganisms-09-00603-f004:**
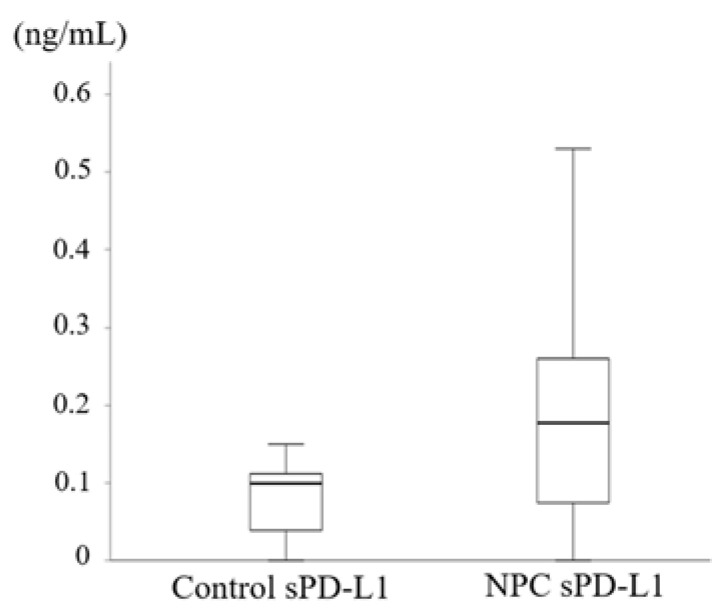
Elevated serum sPD-L1 levels in patients with NPC compared to those in the controls. There was a significant difference in sPD-L1 levels between the controls and the patients with NPC (*p* = 0.031). *p*-value was calculated using the Mann–Whitney U-test. NPC: nasopharyngeal carcinoma, sPD-L1: soluble PD-L1.

**Figure 5 microorganisms-09-00603-f005:**
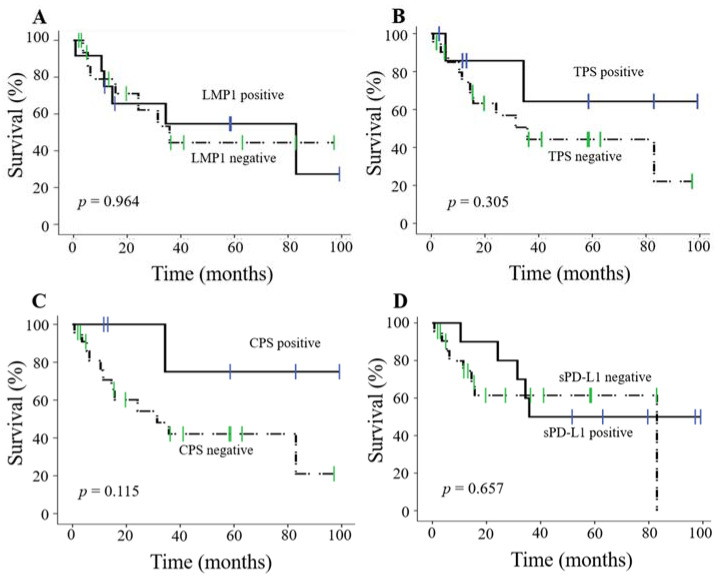
Kaplan–Meier curves of progression-free survival (PFS) in patients with NPC. The *p*-value was calculated using log-rank test. (**A**) Kaplan–Meier curves of LMP1-positive (≥LMP1 expression score 10%) and LMP1-negative (<LMP1 expression score 10%) samples. (**B**) Kaplan–Meier curves of PD-L1 TPS-positive (≥10%) and PD-L1 TPS-negative (<10%) samples. (**C**) Kaplan–Meier curves of PD-L1 CPS-positive (≥10%) and PD-L1 CPS-negative (<10%) samples. (**D**) Kaplan–Meier curves of sPD-L1-positive (≥0.1 ng/mL) and sPD-L1-negative (<0.1 ng/mL) samples. * *p* < 0.05. LMP1: latent membrane protein 1, TPS: tumor proportion score, CPS: combined positive score, sPD-L1: soluble PD-L1.

**Table 1 microorganisms-09-00603-t001:** Characteristics of 35 NPC patients.

Characteristics	All Patients (*n* = 35)
Age (years ± SD)	55.8 ± 12.5
Gender (%)	
Male	32 (91)
Female	3 (9)
Smoking (%)	
Never	9 (26)
Past and present	26 (74)
Alcohol (%)	
Never	15 (43)
Past and present	20 (57)
Tumor stage (%)	
T1–2 (early)	8 (23)
T3–4 (advanced)	27 (77)
Lymph node metastasis (%)	
N0 (negative)	4 (11)
N1–3 (positive)	31 (89)
Metastasis (%)	
M0 (negative)	33 (94)
M1 (positive)	2 (6)
Stage (%)	
I–II (early)	8 (23)
III–IV (advanced)	27 (77)
Histology (WHO Type)	
Keratinizing squamous cell carcinoma	0 (0)
Non-keratinizing carcinomaDifferentiated type	21 (60)
Non-keratinizing carcinomaUndifferentiated type	14 (40)
Recurrent outcomes (%)	
No recurrence	22 (63)
Recurrence	13 (37)
PFS (mean month ± SD)	34.3 ± 28.5
OS (mean month ± SD)	49.5 ± 31

NPC, nasopharyngeal carcinoma; SD, standard deviation; PFS, progression-free survival; OS, overall survival.

**Table 2 microorganisms-09-00603-t002:** Characteristics of 32 patients with NPC according to LMP1 and PD-L1 (TPS and CPS) expression.

Characteristics	All Patients (*n* = 32)	Tumor Proportion Score (%)	*p-*Value	Combined Positive Score (%)	*p-*Value	LMP1 Expression Score (%)	*p-*Value
		(Mean ± SD)		(Mean ± SD)		(Mean ± SD)	
Age (years ± SD)	55.4 ± 12.9		0.458		0.589		0.734
<50	40 ± 10.5	19 ± 18		32 ± 29		20 ± 15	
≥50	62.4 ± 6.4	26 ± 22		37 ± 26.3		20 ± 24	
Gender (%)			0.624		0.624		0.009 *
Male	29 (91)	23 ± 20.4		34 ± 26.4		18 ± 19.9	
Female	3 (9)	30 ± 19		45 ± 24.5		46 ± 9.9	
Smoking (%)			0.967		0.934		0.711
Never	9 (28)	23 ± 15.2		37 ± 24.5		19 ± 19.8	
Past and present	23 (72)	24 ± 22.1		35 ± 27.1		21 ± 21.3	
Alcohol (%)			0.675		0.326		0.675
Never	13 (41)	21 ± 15.7		30 ± 21.1		21 ± 16.8	
Past and present	19 (59)	26 ± 22.9		39 ± 28.9		20 ± 23.3	
Tumor stage (%)			0.94		0.85		0.323
T1–2 (early)	19 (59)	23 ± 18.9		36 ± 26.8		22 ± 17.4	
T3–4 (advanced)	13 (41)	26 ± 22.3		34 ± 25.8		18 ± 25	
Lymph node metastasis (%)			0.64		0.527		0.891
N0 (negative)	4 (12)	19 ± 13.9		27 ± 18.7		20 ± 19.5	
N1–3 (positive)	28 (88)	25 ± 21.1		36 ± 27.1		20 ± 21.1	
Metastasis (%)			0.79		0.847		0.907
M0 (negative)	30 (94)	24 ± 20.4		35 ± 26.7		21 ± 21.5	
M1 (positive)	2 (6)	26 ± 20.5		33 ± 21		15 ± 7	
Stage (%)			0.03 *		0.061		0.656
I-II (early)	7 (22)	10 ± 12.9		18 ± 19.8		22 ± 13.9	
III-IV (advanced)	25 (78)	28 ± 20.3		40 ± 26		20 ± 22.5	
Recurrent outcomes (%)			0.289		0.578		0.107
No recurrent	19 (59)	21 ± 22.4		32 ± 28.7		23 ± 22.5	
Recurrent	13 (41)	29 ± 16		40 ± 21.9		16 ± 17.5	

* *p* < 0.05. NPC: nasopharyngeal carcinoma, SD: standard deviation.

**Table 3 microorganisms-09-00603-t003:** Characteristics of 32 patients with NPC according to sPD-L1 levels.

Characteristics	All Patients (*n* = 32)	sPD-L1 (ng/mL) (Mean ± SD)	*p*-Value
Age (years ± SD)	54.6 ± 12.3		0.251
<50	39.9 ± 10	0.15 ± 0.08	
≥50	62.2 ± 6	0.16 ± 0.12	
Gender (%)			0.855
Male	29 (91)	0.18 ± 0.13	
Female	3 (9)	0.16 ± 0.11	
Smoking (%)			0.145
Never	9 (28)	0.23 ± 0.12	
Past and present	23 (72)	0.16 ± 0.13	
Alcohol (%)			0.562
Never	13 (41)	0.2 ± 0.17	
Past and present	19 (59)	0.16 ± 0.1	
Tumor stage (%)			0.036 *
T1–2 (early)	18 (56)	0.14 ± 0.13	
T3–4 (advanced)	14 (44)	0.19 ± 0.13	
Lymph node metastasis (%)			0.952
N0 (negative)	3 (9)	0.16 ± 0.13	
N1–3 (positive)	29 (91)	0.18 ± 0.14	
Metastasis (%)			0.847
M0 (negative)	30 (94)	0.18 ± 0.14	
M1 (positive)	2 (6)	0.16 ± 0.02	
Stage (%)			0.514
I–II (early)	6 (19)	0.14 ± 0.13	
III–IV (advanced)	26 (81)	0.19 ± 0.13	
Recurrent outcomes			0.431
No recurrent	19 (59)	0.18 ± 0.13	
Recurrent	13 (41)	0.17 ± 0.14	

* *p* < 0.05. NPC, nasopharyngeal carcinoma; sPD-L1, soluble PD-L1; SD, standard deviation.

**Table 4 microorganisms-09-00603-t004:** Cox proportional hazard regression analysis in the 32 patients with NPC.

Characteristics	Univariate Analysis		Multivariate Analysis	
	Hazard Ratio (95%CI)	*p-*Value	Hazard Ratio (95%CI)	*p-*Value
Gender (male)	3.617 (0.758–17.272)	0.107		
Smoking (smoking)	0.328 (0.109–0.985)	0.047 *	0.224 (0.068–0.743)	0.014 *
Alcohol (drinking)	0.355 (0.106–1.187)	0.093		
Tumor (T3–4)	0.825 (0.275–2.477)	0.732		
Lymph node (positive)	1.707 (0.215–13.546)	0.613		
Metastasis (positive)	44.983 (4.001–505.715)	0.002 *	86.961 (6.936–1090.311)	0.001 *
Stage (III–IV)	27.608 (0.050–15205.502)	0.303		
PD-L1 TPS (positive)	2.162 (0.478–9.775)	0.316		
PD-L1 CPS (positive)	4.474 (0.580–34.500)	0.151		
LMP1 (positive)	1.026 (0.341–3.082)	0.964		
sPD-L1 (≥0.1 ng/mL)	1.293 (0.415–4.031)	0.658		

** p* < 0.05. NPC: nasopharyngeal carcinoma, CI: confidence interval, LMP1: latent membrane protein 1, TPS: tumor proportion score, CPS: combined positive score, sPD-L1: soluble PD-L1.

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
