# Peer review of "Epstein–Barr Virus LMP1 Induces Soluble PD-L1 in Nasopharyngeal Carcinoma"

_microorganisms, 2021, doi:10.3390/microorganisms9030603_

Round 1

Reviewer 1 Report

This is an interesting study about the role of Epstein-Barr virus LMP1 in inducing soluble PD-L1 in nasopharyngeal carcinoma. By means of in vitro and in vivo studies, the authors found that LMP1 induces both sPD-L1 and PD-L1, 29 which are associated with NPC progression.

The paper is well written. However, some issues remain.

In the Materials and methods section, the authors should specify how were chosen the 10 control patients. Did the authors excluded subjects with pathologies that may interfere with sPD-L1 levels?

Have the 35 patients received some treatments before the biopsies and serum collection, used for this study? If chemotherapy or radiotherapy were administered before collecting the samples, it may be a bias. Moreover, the authors must specify if any sample derived from a tumor recurrence.

Please insert the paragraphs on western blotting (2.5) before the description of serum samples (2.4).

Since 3 patients had a double cancer, they should be removed from survival analyses, in order to avoid a bias.

The authors must describe how they induced the expression of LMP1 in the non-malignant EBV-negative nasopharyngeal cell line NP69T.

In Table 2 and Figure 5, please better specify that TPS and CPS refers to PD-L1 and not to LMP1.

Author Response

Thank you very much for reviewing our manuscript and offering valuable advice. We have addressed your comments with point-by-point responses and revised the manuscript accordingly.

Point 1: In the Materials and methods section, the authors should specify how were chosen the 10 control patients. Did the authors excluded subjects with pathologies that may interfere with sPD-L1 levels?

Response 1:

Previous studies have reported that sPD-L1 levels are upregulated in various malignancies, infectious disease, and autoimmune diseases. Thus, we selected control patients without pathologies that may affect sPD-L1 levels. Now, we have rewritten the Materials and methods section, as follows.

Page 3, Line 109; Control samples were obtained from 10 patients who did not have any pathologies that may affect sPD-L1 levels, including carcinomas, infectious disease, and autoimmune diseases at diagnosis.

Point 2: Have the 35 patients received some treatments before the biopsies and serum collection, used for this study? If chemotherapy or radiotherapy were administered before collecting the samples, it may be a bias. Moreover, the authors must specify if any sample derived from a tumor recurrence. 

Response 2: The 35 patients were with primary NPC and all patients did not receive any treatments before the biopsies and serum collection. All samples were not derived from a tumor recurrence. We have added this information in the Materials and methods section, as follows.

Page 2, Line 66; All patients did not receive any treatments before the biopsies and all samples were not derived from a tumor recurrence.

Page 3, Line 108; All patients did not receive any treatments before the serum collection and all samples were not derived from a tumor recurrence.

Point 3: Please insert the paragraphs on western blotting (2.5) before the description of serum samples (2.4).

Response 3: We have inserted the paragraphs on western blotting before the description of serum samples.

Point 4: Since 3 patients had a double cancer, they should be removed from survival analyses, in order to avoid a bias.

Response 4: I appreciate for your suggestion. We have excluded 3 patients who had a double cancer. We have reanalyzed survival analyses and changed the Figure 5, Table 4, Figure S1, and Table S1. In addition, we have added this information in the Materials and methods section, as follows.

Page 3, Line 123; We excluded 3 patients who had a double cancer from survival analyses.

Point 5: The authors must describe how they induced the expression of LMP1 in the non-malignant EBV-negative nasopharyngeal cell line NP69T.

Response 5: The LMP1 gene was introduced into NP69T by retroviral vector pLNSX-LMP1. We have added this information in the Materials and methods section, as follows.

Page 2, Line 60; Human immortalized nasopharyngeal cell lines, NP69T and NP69T-LMP1 cells (NP69T cells transfected with pLNSX-LMP1 and stably expressing LMP1), were a kind gift of Dr. George Sai Wah Tsao (University of Hong Kong) [12]. These cell lines were cultured in keratinocyte serum-free medium (Thermo Fisher Science, Kyoto, Japan).

Point 6: In Table 2 and Figure 5, please better specify that TPS and CPS refers to PD-L1 and not to LMP1.

Response 6: I appreciate for your suggestion. We have added the information that TPS and CPS refers to PD-L1 expression in Table 2 and Figure 5.

Reviewer 2 Report

The authors investigated whether EBV-positive NPC patients had increased levels of the sPD-L1 protein. In a relatively small cohort of 35 patients they found that sPD-L1 serum levels were higher in the NPC patients than in the control group, however no correlation with prognosis was observed.

The study despite having some limitations is fairly well designed, however the following comment can be addressed to improve the quality:

  • Figure 1: it would be valuable to add here also sPD-L1 levels in EBV infected vs uninfected cell NPC cell lines since ectopic expression of LMP1 may not reflect what would happen in the context of whole viral infection.
  • Figure 2: would it be possible to do co-staining for LMP1 and sPD-L1 on the same tumor section?

Author Response

Thank you very much for reviewing our manuscript and offering valuable advice. We have addressed your comments with point-by-point responses and revised the manuscript accordingly.

Point 1: it would be valuable to add here also sPD-L1 levels in EBV infected vs uninfected cell NPC cell lines since ectopic expression of LMP1 may not reflect what would happen in the context of whole viral infection.

Response 1: I agree with your suggestion. A previous study reported PD-L1 expression in EBV-positive NPC cell line, C666-1, was significantly higher than that in EBV-negative cell lines and their data showed that EBV infection was associated with up-regulation of PD-L1 cellular expression. We hypothesize that EBV infection also enhanced with sPD-L1 (secreted form of PD-L1) levels. However, we do not have EBV-positive cell lines right now. In future, we plan to analyze whether infection of recombinant EBV to EBV-negative nasopharyngeal cell line affects sPD-L1 level.

Point 2: would it be possible to do co-staining for LMP1 and sPD-L1 on the same tumor section?

Response 2: I appreciate for your suggestion. We did dual fluorescence immunostaining of LMP1 and PD-L1. There were several cells that expressed both LMP1 and PD-L1 in the NPC tissue sample. We have added these figures in Figure 2. In addition, we have added this information in the Materials and methods section and the Result section, as follows.

Page 2, Line 80; Furthermore, 7 of 32 primary NPC specimens were used for dual fluorescence immunostaining of PD-L1 and LMP1. Paraffin sections were deparaffinized, treated with 3% hydrogen peroxide, and incubated with a protein blocker (Dako, Glostrup, Denmark). The sections were incubated at 4 °C overnight with primary antibodies. Next, the sections were exposed to goat anti-mouse Alexa Fluor 594 and anti-rabbit Alexa Fluor 488 IgG secondary antibodies (1:500, Thermo Fisher Science, Kyoto, Japan) after washing with PBS. Then, the sections were counterstained with 4′,6-diamidino-2-phenylindole (DAPI, P36962, Thermo Fisher Scientific, Kyoto, Japan).

Page 4, Line 156; Next, we performed dual fluorescence immunostaining and assessed the expression of LMP1 and PD-L1 in the same tissue sample. There were several cells that expressed both LMP1 and PD-L1 in the NPC tissue sample (Figure 2F).

Round 2

Reviewer 1 Report

Thanks you for improving the manuscript.

Reviewer 2 Report

The authors have addressed my points.

I can now recommend the study for publication.